# Effect of Antioxidant Supplementation on Milk Yield and Quality in Italian Mediterranean Lactating Buffaloes

**DOI:** 10.3390/ani12151903

**Published:** 2022-07-26

**Authors:** Chiara Evangelista, Umberto Bernabucci, Loredana Basiricò

**Affiliations:** Department of Agricultural and Forests Sciences (DAFNE), University of Tuscia, 01100 Viterbo, Italy; chiara.evangelista@unitus.it (C.E.); basiri@unitus.it (L.B.)

**Keywords:** dairy buffalo, milk, antioxidant, clotting properties

## Abstract

**Simple Summary:**

Buffalo milk has good dairy properties that allow its transformation into excellent cheeses, in particular mozzarella. What greatly influences the quality of buffalo milk, and its properties is the diet administered; for this reason, we wanted to test whether a supplement of antioxidants (SOD, Zn, and Se) in the diet of lactating buffaloes could affect the quantity and quality of their milk. The experiment was carried out on sixty-six lactating buffaloes on a farm in central Italy. Individual milk samples were collected every four weeks at morning and evening milking. The results obtained showed that treatment had no effect on feed intake, feeding behaviour, and feed efficiency and improved milk yield and milk clotting properties (MCP).

**Abstract:**

Buffaloes are raised mainly to obtain milk that is nutritionally very rich. The technological characteristics of buffalo milk are optimal for processing into cheese, and it is mainly used to produce mozzarella cheese. Under stressful conditions, buffaloes, like other animals, produce milk qualitatively poorly. The stressors that can affect the quality of production are, in addition to other factors, deficiencies in nutrients such as vitamins, antioxidants, and minerals. In this study, we evaluated the effect of antioxidant supplementation on the quality of buffalo milk. Sixty-six buffaloes were enrolled and subdivided into two balanced groups of 33 each. The ZnSe group received 0.2 kg/head/day of Bufalo Plus^®^ containing antioxidants and barley meal, CaCO_3_ and MgCO_3_ mix; the control group was supplemented with 0.2 kg/head/day of barley meal, CaCO_3_ and MgCO_3_ mix. The two groups were fed ad libitum with a total mixed ration (TMR). The amount of diet distributed was recorded daily, and the residue in the trough manger was recorded three times per week. TMR samples were taken every two weeks for each group. Daily milk yield was recorded twice a week. Milk samples were collected every four weeks and analysed for chemical and technological properties. Furthermore, milk total antioxidant capacity was determined. The results obtained showed that the antioxidant supplement had no effect on feed intake, feeding behaviour, and feed efficiency. The treatment positively influenced milk production while it did not affect the chemical characteristics of the milk. In addition, the supplement of antioxidants improved the milk clotting properties (MCP). The supplement did not affect the antioxidant activity of the milk.

## 1. Introduction

Buffalo milk is a very nutrient-rich milk containing a high percentage of both fat and protein and has double the energy content and lower cholesterol levels (6.5 mg/100 g milk) than cow’s milk [1]. It is a good source of vitamins A, D, C, and B6 and minerals. It is very rich in calcium, magnesium, phosphorus, potassium, zinc [2], and iron [3] and has more conjugated linoleic acid (CLA) compared to cow’s milk [4]. Buffalo milk, being very rich in nutrients, has a strong tendency to be transformed into cheese and has a high cheese yield relative to cow’s milk [5]. The high concentration of nutrients, on the one hand, is particularly interesting for the cheese yield; on the other hand, it is an aspect that potentially represents a drawback for the development of microorganisms, which use the high nutritional content to develop, worsening the nutritional characteristics and hygiene of the milk.

Milk and milk products are nutritious food items containing numerous essential nutrients such as oleic acid, conjugated linoleic acid, omega-3 fatty acids, vitamins, minerals, and bioactive compounds such as antioxidants [6]. Antioxidants are chemical substances that scavenge/neutralise free radicals, and foods should contain enough of a concentration of antioxidants to prevent oxidative stresses (OS) [7]. OS is defined as an imbalance between the production of free radicals and antioxidant defences in favour of the former and is closely related to the immune system and inflammatory status [8]. The antioxidant capacity of milk and dairy products is due to the amino acids containing sulphuric acids such as cysteine, vitamins A, E, carotenoids, enzyme systems, superoxide dismutase, catalase, and glutathione peroxidase. Fresh buffalo milk has a greater antioxidant capacity compared with fresh cow’s milk [7] because buffalo milk has a higher concentration of many antioxidant compounds such as vitamin E (5.5 mg/100 mL in buffalo milk and 2.1 mg/100 mL in cow milk) or vitamin C (3.66 mg/100 mL in buffalo milk and 0.94 mg/100 mL in cow milk) [9] and other antioxidant compounds such as selenium (Se), zinc (Zn), tyrosine, cysteine, and catalase; furthermore, catalase activity is 2–4 times higher in buffalo milk as compared with cow milk.

There is a range of dietary antioxidants that can be provided in feed, which include vitamin E, carotenoids, polyphenolics, antioxidant enzymes, Zn, and Se (as a precursor to selenoproteins). Zn, Se, and Cu supplementations have been associated with the increased antioxidant capacity of superoxide dismutase (CuZnSOD), glutathione peroxidase (GSH-Px), and serum ceruloplasmin (CP), respectively, resulting in reduced milk somatic cell counts (SCC) [10]. Zinc and Se are essential for the health and performance of all animal species because they are involved in several aspects of cellular metabolism [11]. There are several publications showing that trace mineral supplementation can affect the health, reproductive status, immune function, and lactation performance of dairy cows [12,13,14]. Weiss et al. [15] reported that Zn, Cu, and Se are needed to maintain the antioxidant activity of dairy cows’ immune system, in particular the defensive reactions of the mammary gland against mastitis.

The use of trace minerals from organic sources in animal nutrition can be an important tool for increasing milk production and maintaining the health status of animals [16]. However, the literature on the role of micronutrient and antioxidant supplementation in the diet of buffaloes and its effect on the production and quality of milk is very limited.

Therefore, this study aimed to evaluate the effect of antioxidant supplementation on milk yield and quality of lactating dairy buffaloes.

## 2. Materials and Methods

### 2.1. Animals and Feeding

Sixty-six healthy Mediterranean lactating buffaloes were subdivided into two groups of thirty-three animals balanced by parity, days in milk (DIM), and daily milk yield (Table 1). The two groups were separated by gates, but both were in the same building with the same housing conditions (Appendix A).

The two groups were the ZnSe group and control group. The ZnSe group was fed with a standard diet supplemented with antioxidants (treated group); the control group was fed with a standard diet. The composition of the diet distributed is shown in Table 2. The ZnSe group received 0.2 kg/head/day of Bufalo Plus^®^ (SIVAM, Italy) added to TMR. The supplement (Bufalo Plus^®^) composition was the following: CaCO_3_, MgCO_3_, barley meal, Melofeed^®^ (rich in the antioxidant enzyme superoxide dismutase, SOD), hydrated amino acid zinc chelate-3b606 (4000 mg/kg), selenium zinc-L-selenomethionine-3b818 (16 mg/kg), Ca (20.5%), and Mn (11%). The control group was supplemented with 0.2 kg/head/day of barley meal, CaCO_3_, and MgCO_3_ mix.

The experimentation started in early May and ended in mid-August and lasted 103 days.

The buffaloes were milked two times a day in a herringbone milking parlour with 11 stalls, equipped with measuring jars with the possibility to measure milk yield (L) and take individual milk samples.

The two groups were fed *ad libitum* with a total mixed ration (TMR), distributed once a day at 7:30 a.m. with a two-augers vertical mixer wagon. The amount of diet distributed was recorded daily. The residue in the trough manger was recorded three times a week.

TMR samples were taken every two weeks for each group. A total of 8 TMR samples per group were collected.

The TMR was analysed for physical and chemical characteristics. Samples were sieved using the 3-screen (19, 8, and 4 mm) Penn State Particle Separator (PSPS; [17]) for particle size determination. For chemical analysis, dry matter (DM) was measured after oven drying at 65 °C to constant weight. Then, TMR samples were ground through a mill (Retsch Müller, Germany) to pass a 1 mm screen. To maintain optimal preservation conditions, sealed polyethylene containers were used to store prepared samples. Samples were analysed for crude protein [18], ash [19], ether extract [20], and starch [18] using a K-TSTA assay kit (Megazyme International, Bray, Ireland), and neutral detergent fibre (aNDF), acid detergent fibre (ADF), and acid detergent lignin (ADL) were analysed using an Ankom200 Fiber Analyzer (Ankom Technology, Macedon, NY) according to Van Soest et al. [21]. TMR was also analysed on site for determining homogeneity index (H.I.) and selection index (S.I.) at 1.5 h from distribution, with portable NIRS (PoliSPEC^NIR^ diode array spectrometers with a spectral range from 902 to 1680 nm, ITPhotonics, Breganze (VI), Italy).

All data are reported as percentages on a dry matter basis, except for H.I. and S.I., which were expressed according to procedure indicated by the manufacturer.

### 2.2. Milk Yield and Analysis

Individual daily milk yield was recorded twice a week. Milk samples were collected every four weeks at morning and evening milking, directly from the measuring jars attached to the milking machine and packaged in 50 mL plastic tubes containing Bronopol^®^ (2-Bromo-2-Nitropropane-1,3-Diol) as a preservative. Milk samples were refrigerated (4 °C) and analysed within 24 to 36 h after collection. Content (%) of fat, protein, lactose and solid not fat (SNF), and pH were determined by F.T.I.R. spectrophotometry, using a MilkoScanTM 7 RM (Foss Analytics, Hillerød, Denmark) calibrated with appropriate buffalo milk standards. Somatic cell counts (SCC, thousand cells/mL) were determined by fluoro-optoelectronics method using a FossomaticTM FC (Foss Analytics, Hillerød, Denmark). To achieve a normal distribution of the SCC, the somatic cell score (SCS, units) was calculated as follows [22]:SCS = log_2_ (SCC/100,000) + 3(1)

The corrected milk of the total daily milk was converted into buffalo energy-corrected milk (ECM) using the following formula [23]:Y = {[(F × MY − 40) + (P × MY − 31)] × 0.01155 + 1} * MY(2)
where Y is the quantity (kg) of ECM equivalent to 1 kg of milk produced; MY is the milk yield; F is the percentage of fats; P is the percentage of proteins.

Feed efficiency ratio (FEr) was calculated relating the ingestion of DM, in the days before and after milk sampling, to the ECM.

Milk clotting properties (MCP) (RCT: rennet coagulation time, min; k_20_: curd firming time, min; a_30_: curd firmness, mm) were determined by using a Formagraph instrument (Mape System, Firenze, Italy) according to Zannoni and Annibaldi [24]. For MCP determination, milk samples (10 mL) were heated to 35 °C, and 200 µL of commercial calf rennet (Clerici s.p.a.-Sacco s.r.l., Cadorago, Italy; 75% chymosin and 25% bovine pepsin; 175 international milk clotting units/mL; strength 1:15.000 in Italian commercial units) diluted to 1% (wt/wt) in distilled water was added to the milk samples. Milk clotting properties were determined over the next 30 min after the addition of calf rennet.

Total antioxidant activity (TAA) of buffalo milk was determined using the Oxy Adsorbent Test (Diacron International, Grosseto, Italy) and a spectrophotometric plate reader (Sunrise Plate Reader, Tecan Trading AG, Männedorf, Switzerland) at 540 nm wavelength. The test is used to evaluate serum or plasma antioxidant barriers and for the evaluation of milk antioxidant capacity [25,26]. Antioxidant capacity was assessed according to the producer’s instructions, with slight modifications. The antioxidant activity of the milk was evaluated by measuring the ability of milk samples to resist the great oxidant action of hypochlorous acid (HClO), a potent and physiological oxidant. All antioxidant molecules in milk samples, scavengers, and shock absorbers related to neutralisation of free radicals were measured with this assay. Briefly, the sample of buffalo milk was incubated with an excess of HClO solution, and then the residual HClO reacted with a chromogenic mixture and developed a coloured complex. The measured optical densities were proportional to the concentration of HClO and indirectly associated with the antioxidant capacity of samples. TAA values were expressed as µmol of HClO/mL, according to the formula reported in the producer’s instructions. For the assay, whole buffalo milk samples were diluted 1:75 (*v*/*v*) with distilled water.

### 2.3. Statistical Analysis

A nested ANOVA with a mixed-effects model (an extension of one-way ANOVA in which each group is subdivided into subgroups) was used to test for a significant effect of “controls”, “groups”, and “samples” (i.e., factors) on the measured parameters. The analyses were then performed by contrasting each parameter against each factor and their interactions by using “animal” as a random variable in the models. A multiple pairwise comparison test was performed by using a Bayes factor analysis between group levels with corrections for multiple testing. All analyses were performed by using the function “lme” and “pairwise_comparisons” in the packages “nlme” [27] and “pairwiseComparisons” [28] of R [29]. Results are expressed as Lsmeans ± SE, and significance was declared at *p* < 0.05.

## 3. Results

### 3.1. Feed Intake and Diet Characteristics

Changes to feed intake during the experimental period are reported in Figure 1. The average amount of feed intake (Lsmean ± SE), as it is, was 27.97 ± 1,55 vs. 27.58 ± 1.76 Kg/head/day for the ZnSe group and control group, respectively (Figure 1a). Dry matter intake (Figure 1b), on average (Lsmean ± SD), was 16.65 ± 0.73 vs. 16.52 ± 0.88 Kg/head/day in the ZnSe group and control group, respectively. Both feed intake and dry matter intake did not differ between the two groups.

The average composition of the diets distributed to the two groups is shown in Table 3. Both chemical composition and physical characteristics were not different between the two diets.

Table 4 shows the results on the H.I. and S.I. of the TMR distributed to the two groups. According to the classification scale of homogeneity, the H.I. of the TMR was classified as “sufficiently homogeneous” (50 < HI < 65). The homogeneity of TMR distribution was not different between the two groups. As regards the selection index at 1.5 h after TMR distribution, buffalo performed a “modest selection” (20 < SI <40). Moreover, the selection index was not different between the two groups. This means that the preparation and distribution of rations were identical for the two groups.

### 3.2. Milk Yield and Milk Quality Traits

#### 3.2.1. Milk Yield

The trend of milk yield (Lsmeans ± SE) during the experimental period (L/head/day) is shown in Figure 2a. Figure 2b shows the overall average (Lsmeans ± SE) daily milk yield. Figure 2c reports the total milk yield (Lsmeans ± SE) in the two groups estimated by calculating the area under the lactation curve. The average daily milk yield was greater (*p* < 0.001) in the ZnSe group compared with the control group (9.12 ± 0.02 vs. 8.81 ± 0.02 L/head/day for the ZnSe group and control group, respectively) with an improved milk production of 0.31 L/head/day. Moreover, the total milk yield was greater (994.93 ± 32 vs. 964.99 ± 44 L/head; *p* < 0.001) in the ZnSe group compared to the control group. During the experimental period, the ZnSe group produced 30 L/head more than the control group.

#### 3.2.2. Milk Quality Traits

The results (Lsmeans ± SE) on milk quality traits and SCS and the ECM and feed efficiency ratio are shown in Table 5. There was not any difference for all parameters tested between the two groups (Table 5).

The clotting parameters of milk were positively influenced by treatment (RCT and a_30_ *p* < 0.01 **; k_20_ *p* < 0.05 *). Lsmeans (±SE) of RCT was 15.14 ± 0.85 vs. 16.37 ± 0.85 min, 2.49 ± 0.26 vs. 2.74 ± 0.26 min for k_20_, and 51.31 ± 2.46 vs. 47.94 ± 2.47 mm for a_30_, for the ZnSe group and control group, respectively (Figure 3).

Milk total antioxidant activity (TAA) was not affected by treatment. TAA was 408.15 ± 24.18 vs. 432.97 ± 25.64 µmol of neutralised HClO/mL for the ZnSe group and control group, respectively (Figure 4).

## 4. Discussion

The results of the present study showed that antioxidant supplementation, SOD, Zn, and Se, in the diet of lactating buffaloes positively influenced milk yield and improved the milk clotting properties (MCP). The addition of an organic source of mineral antioxidant supplement did not affect dry matter intake, as well as in other studies carried out in buffaloes [30] and in dairy cows [16], and it had no effect on the chemical characteristics of the milk.

Dry matter intake observed in this study is in line with what was recently reported by Salzano et al. [2] for lactating buffaloes fed by TMR, and it is lower compared with that observed by Tufarelli et al. [31] in a study carried out during spring. This study was carried out during summer and most likely could have reduced feed consumption in both groups. Both chemical composition and physical characteristics were not different between the two diets, and both distribution H.I. and S.I. were not different between the groups. These data indicate that preparation and distribution of diets and feeding behaviour of buffaloes were identical for the two groups.

Daily milk yield agrees with the survey conducted by Costa et al. [32] on a large population of Italian water buffalo over a period of five years (8.81–9.24 kg/head/day), and it is lower than that reported by the A.I.A. statistics [33] (9.65 kg/head/day) referring to buffaloes raised in the Lazio region. Our results are lower than what was reported by Salzano et al. [2]. These differences could be due to different factors such as parity and the stage of lactation of the subjects used in the experiments [34,35].

In the current study, the daily milk yield was improved by antioxidant supplementation (SOD, Zn, and Se). These results agree with findings by Singh et al. [30], who found that the antioxidant supplementation (Cu, Zn, and vitamins A and E) improved milk production in buffaloes. Moreover, Kantwa et al. [36] and Tanwar et al. [37] reported a significant increase in the milk yield of buffaloes supplemented with chelated mineral mixture. Other studies have reported an increase in the milk yield in dairy cows supplied with organic mineral mixtures [38,39] and area-specific mineral mixtures [40].

In agreement with other studies [36,41], the greater milk yield of the treated group compared with the control group might be due to the effect of adding the minerals responsible for the stimulation of basket cells or myoepithelial cells of the udder and, in particular, Zn may improve udder health [38]. Furthermore, Miranda et al. [42] reported that low levels of antioxidants impaired oxidative status in the mammary gland, which was linked to a reduction in the number of mammary gland epithelial cells. Other authors [43] reported a reduction of apoptotic mammary gland epithelial cells after antioxidants supplementation.

Based on the literature and the results of the present investigation, in which an increased production performance was observed, with a comparable dry matter intake between the ZnSe group and control group, it is possible to hypothesise that the antioxidant treatment was able to improve the oxidative status of animals and milk-producing cells in the udder, leading to a more efficient utilisation of energy for milk production.

The content (%) of fat, protein, and lactose between the treatment and control group was not improved by the antioxidant supplementation in the buffaloes’ diet. According to [40], the effect of area-specific mineral mixture supplementation in dairy cattle has positively influenced milk yield, but no significant differences were observed in the milk’s fat, protein, and lactose contents. Moreover, Kellogg et al. [38] showed that the supplementation of Zn-methionine in dairy cows improved the milk yield, and milk composition did not change. In contrast, De Andrade et al. [44] reported that Se supplementation reduced SCC, fat, and protein percentages in buffalo milk. In a study by Singh et al. [30], supplementation of antioxidant micronutrients in buffaloes improved milk fat and protein content and had no effect on the lactose content.

Milk fat content observed in the present study agrees with data reported by Costa et al. [32] and was greater than that reported by A.I.A. [33]. The protein percentage was similar to data reported by A.I.A. [33] and by Costa et al. [32]. Lactose content was lower than in other studies on buffalo milk [32,45] and was higher compared with data reported by Niero et al. [46]. Differences for fat, protein, and lactose between our study and data from the literature are probably due to the different experimental conditions including days of lactation, number of lactations, seasons [35], age, health status, feed intake, concentration of the ration, etc.

Milk pH can be considered within the range of normality for buffalo milk [45,46] and was not affected by treatment. To the best of our knowledge, there is a lack of information about the effects of antioxidant supplementation on the pH of buffalo milk. A recent study [47] where extracts of oregano or green tea, endowed with antioxidant properties, were used as feed additives in Jersey cows during the transition period, reported that only the extract of oregano showed a tendency (0.05 < *p* < 0.10) in the reduction of milk pH compared to the control and green tea extract. Those authors explained the tendency of a lower pH with the lower content of SCC observed in the oregano-treated group compared to the other two groups. 

Milk somatic cell count and milk somatic cell score were not affected by the antioxidant treatment, as already reported in other studies [30,31,38]. Somatic cell count was lower than the values reported by Tripaldi et al. [45] (314 × 10^3^ cells/mL) and by Pasquini et al. [48] for buffaloes bred in the Marche region (from 152.84 ± 25.22 to 199.73 ± 23.43 × 10^3^ cells/mL) and was higher compared with mid-lactating Murrah buffaloes in India [34]. The milk somatic cell score falls within the range reported by Costa et al. [49], which reported a range from 2.71 to 3.11. Differences between the results of the present study and findings from others might be also due to the different analytical methods (opto-fluoro-electronic vs. California Mastitis Test) applied.

The content of somatic cells is affected by the health of the udder, by buffalo productivity, by season, by parity, and by the lactation phase [34,50]. Tripaldi et al. [45] recommended 200 × 10^3^ cells/mL as the limit for early identification of an animal affected by subclinical mastitis. Moreover, any change in environmental conditions, poor management practices, and stressful conditions significantly increases the number of somatic cells in milk. The level of somatic cells observed in the present study indicates good hygiene and proper nutrition and a low risk of pathogen infection in both groups, as well as optimal udder health conditions [51].

The result obtained for ECM agrees with what was reported in a survey by Costa et al. [32] (ECM: 14.36–15.75 Kg/d). Salzano et al. [2] reported ECM values of 17.4 ± 0.4 kg/d due to the higher milk yield and fat percentage compared to data of the present study.

The FEr was not different between the two groups. Our result agrees with that reported by Khattab et al. [52] in Egyptian buffaloes. Those authors found an FEr of 0.57 (with an average production of 8.9 L/head/d and DMI of 15.69 kg/head/d).

The data reported here showed that integration to the treated group with the SOD, Zn, and Se positively affected MCP. Milk clotting properties’ values observed in the present study are similar to those reported by Costa et al. [49] in a study on 99 farms in the Lazio region (RCT = 13.25 ± 4.53 min.; k_20_ = 3.96 ± 2.63 min.; a_30_ = 49.28 ± 10.09 mm). It is well known that a different coagulation time and rennet coagulation properties come from differences in milk composition linked to other factors than merely fat and protein [53]. In fact, the clotting properties of milk are affected by SCC, pH, titratable acidity [54,55,56,57], and the content of some minerals including Ca, P, and Ca/P ratio [53], as well as days in milk [56]. In addition, the total casein content (CN) and the percentages of casein fractions affect them as well. It has been reported that a low concentration of κ-CN and a low proportion of κ-CN relative to the total CN were associated with poor and non-coagulating samples [58,59]. Post-translational modifications of caseins also are important; a higher proportion of glycosylated κ-casein seems to be associated with shorter rennet coagulation times [60,61]. Furthermore, it is demonstrated that MCP are strongly influenced by the season [62,63]; in particular, the hot season makes these characteristics worse, making the milk less suitable to produce cheese. Regarding the rheological properties of milk, in the present study, the treatment significantly influenced the ability of milk to clot. It is not easy to explain the improvement of MCP while also considering the lack of differences for the other milk characteristics.

To date, there is no literature concerning the effects of dietary supplementation with SOD, Zn, and Se on the rennet characteristics of buffalo milk; therefore, this topic needs further investigation. In dairy cows, a supplement of mineral complex (Fe, Mn, Zn, Ca, Mg, P, Na, Cu, Cl, Calodat, Se, K) applied through the animals’ drinking water improved milk yield, protein content, and a_30_ but had no effect on RCT and k_20_ [64].

Some studies in dairy cows indicate that the use of trace minerals from organic sources in animal nutrition is important to maintain the antioxidant activity of the immune system and the health status of animals [15,16]. Faulkner and Weiss [65] also suggested that trace minerals improved rumen fermentation and positively affected the nutrient digestibility. In addition, Pino and Heinrichs [66] reported that, in dairy heifers, the supplementation of organic trace minerals (Cu, Zn, Mn, Se, and Co) had no effect on nutrient digestibility but increased total volatile fatty acids production. Therefore, in the present study, the dietary supplementation with SOD, Zn, and Se likely might have provided a greater available amount of energy to the buffalo’s udder, to be used for daily milk production and probably also to carry out the post-translational modifications of the caseins, which play a role in milk clotting properties.

Integration with SOD, Zn, and Se had no effect on buffalo milk TAA. The lack of antioxidant integration on milk TAA in the treated group compared to the control group might be due to the buffalo’s use of antioxidants supplementation to maintain their health status and to increase milk production. It is particularly difficult to compare the different milk TAA values reported in the literature because they were obtained by means of diverse analytical methodologies, but the antioxidant capacity of buffalo milk is greater than that of milk from other dairy species [67], probably because buffalo milk has a high concentration of many antioxidant compounds [9]. TAA values obtained in this study can be compared to those reported by Beghelli et al. [26], who adopted the same assay to measure milk TAA in different dairy species (goats, ewes, cows, and donkeys) and confirmed that buffalo milk has greater antioxidant capacity than milk from other species.

## 5. Conclusions

The effect of Zn, Se, and SOD supplementation on the productivity and quality of buffalo milk has been investigated. Feed intake, feeding behaviour, and feed efficiency were not affected by the treatment. The treatment positively improved milk yield (+3.1%) and the coagulation properties of the milk, and no differences were found for other milk parameters.

It is not easy to explain the improvement of MCP, but the benefits of an appropriate antioxidant supplementation for improving the cheesemaking properties of milk are therefore of interest to the profits of farmers, since 100% of buffalo milk is used for cheese production. Further research is needed to investigate more in depth the effects of supplementation with Zn, Se, and SOD on milk coagulation properties considering other parameters that are strictly linked with milk coagulation attitude.

## Figures and Tables

**Figure 1 animals-12-01903-f001:**
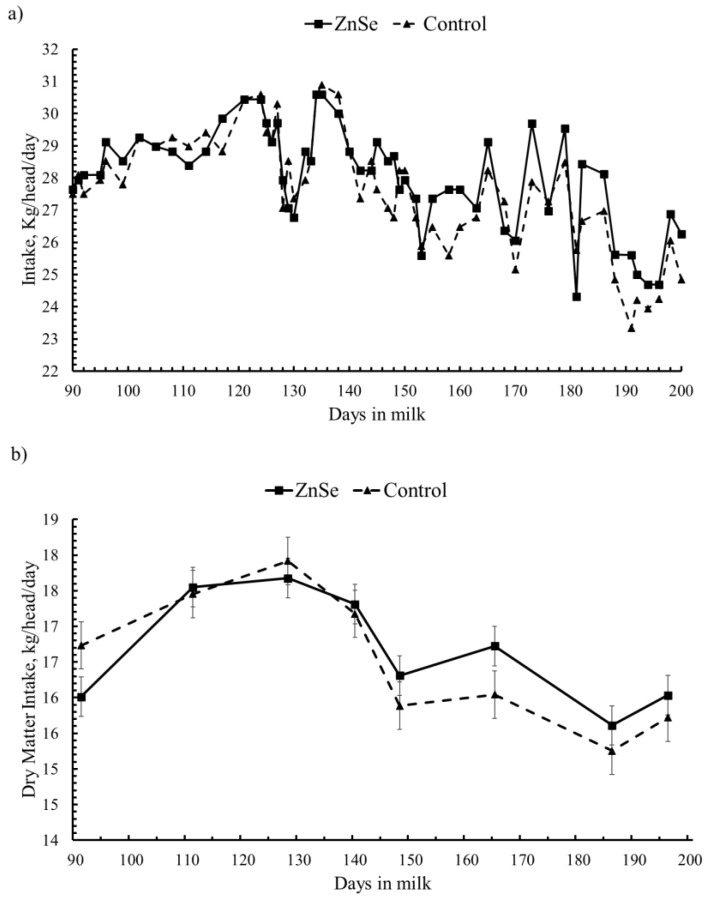
Trend of daily feed intake between the two groups. (**a**) Feed intake; (**b**) dry matter intake. Data are expressed as Lsmeans ± SE.

**Figure 2 animals-12-01903-f002:**
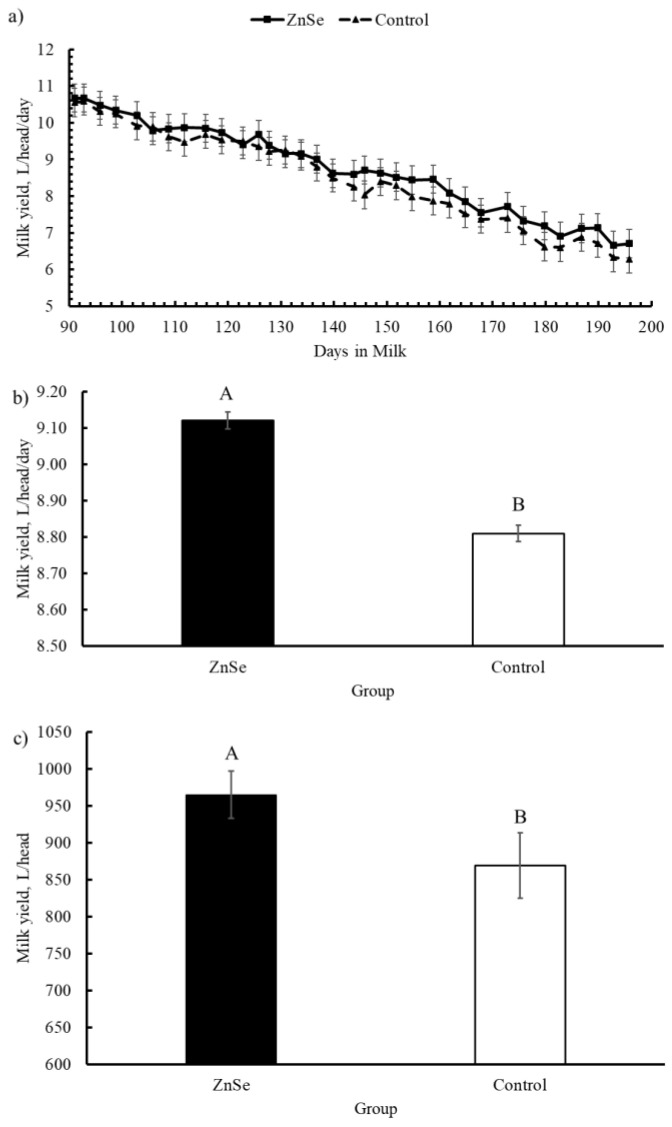
Milk yield trend (L/head/day) for the two groups: (**a**) changes during the trial; (**b**) average milk yield (L/head/day); (**c**) total milk yield (L/head) calculated as area under the curve. Data are expressed as Lsmeans ± SE. A,B = *p* < 0.001.

**Figure 3 animals-12-01903-f003:**
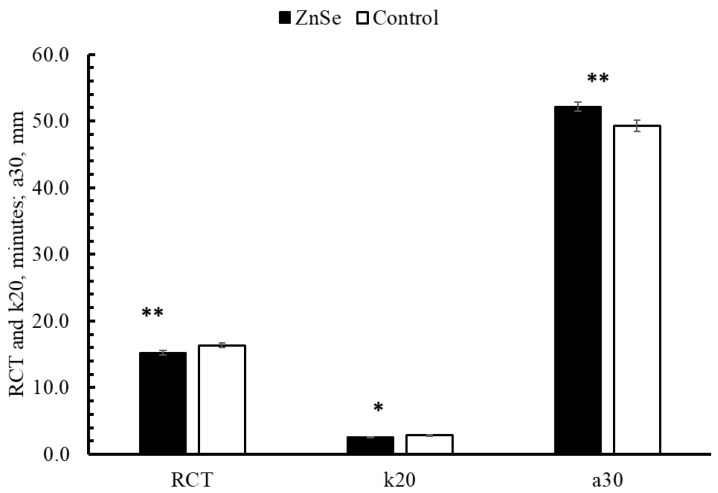
Technological characteristics of milk in the two groups. RCT: rennet coagulation time, min; k_20_: curd firming time, min; a_30_: curd firmness, mm; * *p* < 0.05, ** *p* < 0.01.

**Figure 4 animals-12-01903-f004:**
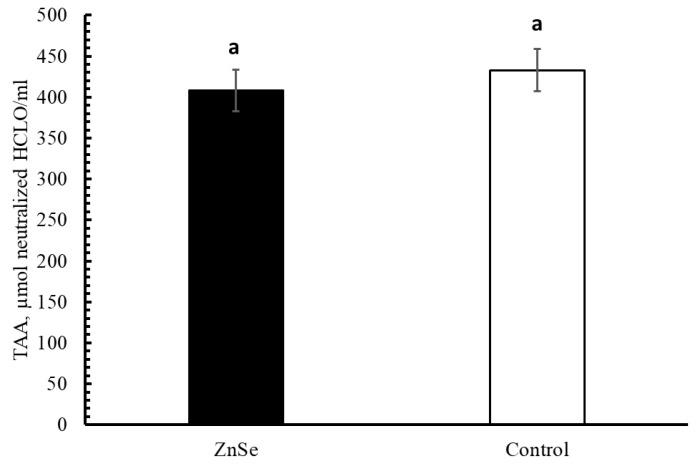
Lsmeans (±SE) of total antioxidant activity (TAA) of milk in the experimental groups. ^a^ Same letters indicate no significant differences.

**Table 1 animals-12-01903-t001:** Characteristics of the two groups at the beginning of the experiment. Data are expressed as means ± SD. ^a^ Same letters indicate no significant differences.

Parameters	ZnSe	Control
Parity (n)	5.2 ± 3.2 ^a^	5.0 ± 3.2 ^a^
Days in milk	91.5 ± 26.7 ^a^	91.2 ± 25.6 ^a^
Daily milk yield (L/head/d)	10.67 ± 2.37 ^a^	10.55 ± 2.16 ^a^

**Table 2 animals-12-01903-t002:** Composition of the standard diet.

Feed	Kg/Head (w.b. ^1^)
Corn silage	10.0
Grass silage	10.0
Commercial mixed feed ^2^	5.0
Alfalfa Hay	4.0
Mixed Hay	3.0
Sodium bicarbonate	0.02

^1^ w.b.: wet basis; ^2^ composition: shelled soybean meal, wheat middling, shelled sunflower meal, corn meal, soy husks, wheat bran, corn gluten meal, barley meal, dried beet pulp, soybean oil, hydrogenated fat, calcium carbonate, sodium bicarbonate, by-products from L-glutamic acid production, sodium chloride, and magnesium oxide. Supplements per kg: vit. A 25,000.00 UI; vit. D3 1800.00 UI; vit. E 30 mg; niacin 50.00 mg; vit. B1 0.40 mg; vit. B2 0.80 mg; vit. B12 0.05 mg; choline chloride 20 mg; trace elements (mg): Fe (20.70); Mn (50.00); Zn (75); Co (0.50); I (0.66); Se (0.25); cupric sulphate (1.00).

**Table 3 animals-12-01903-t003:** Characteristics of total mixed ratio (TMR) distributed to the two experimental groups. Data are expressed as Lsmeans ± SE. ^a^ Same letters indicate no significant differences.

Parameters	ZnSe	Control
Dry Matter, %	59.77 ± 1.7 ^a^	60.20 ± 1.6 ^a^
Ash, %	7.5 ± 0.36 ^a^	7.58 ± 0.48 ^a^
Crude Protein, %	11.24 ± 1.05 ^a^	11.23 ± 0.71 ^a^
Ethereal Extract, %	2.19 ± 0.3 ^a^	2.33 ± 0.52 ^a^
aNDF, %	41.98 ± 2.94 ^a^	42.37 ± 2.26 ^a^
ADF, %	29.48 ± 2.88 ^a^	29.94 ± 2.24 ^a^
ADL, %	3.81 ± 0.7 ^a^	4.10 ± 0.43 ^a^
Starch, %	20.23 ± 1.34 ^a^	20.13 ± 0.83 ^a^
S1, %	10.20 ± 4.23 ^a^	10.37 ± 3.72 ^a^
S2, %	29.95 ± 4.49 ^a^	29.55 ± 4.31 ^a^
S3, %	35.61 ± 1.38 ^a^	34.74 ± 1.68 ^a^
Bottom, %	24.24 ± 3.56 ^a^	25.33 ± 3.48 ^a^

DM: dry matter; ash; crude protein; ethereal extract; aNDF (neutral detergent fibre); ADF (acid detergent fibre); ADL (acid detergent lignin); and starch are on DM basis; S1 = % of ration retained by a sieve with holes of 19 mm; S2 = % of the ration retained by a sieve of 8 mm; S3 = % of the ration retained by a sieve of 4 mm; bottom = % of ration with dimensions <4 mm.

**Table 4 animals-12-01903-t004:** Homogeneity (H.I.) and selection (S.I.) indices of total mixed ratio (TMR) distributed to the two experimental groups. Data are expressed as Lsmeans ± SE. ^a^ Same letters indicate no significant differences.

Parameters	ZnSe	Control
H.I.	55.42 ± 4.30 ^a^	50.12 ± 7.03 ^a^
S.I.	38.17 ± 2.95 ^a^	39.78 ± 4.25 ^a^

**Table 5 animals-12-01903-t005:** Results of milk quality traits, energy-corrected milk (ECM), and feed efficiency ratio (FEr) between the two groups under examination. Data are expressed as Lsmeans ± SE. ^a^ Same letters indicate no significant differences.

Parameters	ZnSe	Control
Fat, %	8.20 ± 1.48 ^a^	8.33 ± 1.61 ^a^
Protein, %	4.64 ± 0.35 ^a^	4.64 ± 0.47 ^a^
Lactose, %	4.66 ± 0.20 ^a^	4.66 ± 0.19 ^a^
Solid Not-fat, %	10.00 ± 0.35 ^a^	10.03 ± 0.50 ^a^
pH	6.63 ± 0.14 ^a^	6.61 ± 0.17 ^a^
SCC, thousand cells/mL	118.78 ± 201.75 ^a^	131.62 ± 188.41 ^a^
SCS	2.76 ± 0.78 ^a^	2.73 ± 0.97 ^a^
ECM, Kg/h/d	14.34 ± 1.10 ^a^	13.64 ± 1.06 ^a^
FEr	0.57 ± 0.09 ^a^	0.56 ± 0.08 ^a^

ECM: energy-corrected milk; SCC: somatic cell count; SCS: somatic cell score = log_2_(SCC/100,000) + 3; FEr: feed efficiency ratio.

## Data Availability

The data presented in this study are available on request from the corresponding author.

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
