# Peer review of "Effect of Antioxidant Supplementation on Milk Yield and Quality in Italian Mediterranean Lactating Buffaloes"

_animals, 2022, doi:10.3390/ani12151903_

Round 1

Reviewer 1 Report

Major focus of introduction section is related to nutritional value of buffalo milk, it should be focused on nutritional characterization and application of ZnSe in foods in the light of relevant literature review. Authors should describe  a mechanism of fat decline in experimental group. Kindly develop correlation between ZnSe and MCP of milk. Letters should be provided in Tables to know the significant difference. 

Author Response

We thank the reviewer for her/his effort in giving suggestions and comments that were useful for improving the manuscript.

R1

R1: Major focus of introduction section is related to nutritional value of buffalo milk, it should be focused on nutritional characterization and application of ZnSe in foods in the light of relevant literature review.

AU: We have added the more info as requested.

R1: Authors should describe  a mechanism of fat decline in experimental group.

AU: Fat content was not different between groups as described into the manuscript (see L…. and table 5)

R1: Kindly develop correlation between ZnSe and MCP of milk.

AU: We did not measure Zn and Se in milk, so, we cannot calculate the correlations between those elements and MCP in milk.

R1: Letters should be provided in Tables to know the significant difference.

AU: Actually, letters were not reported because there were not any differences. Anyway, we reported the letter in tables.

Reviewer 2 Report

The article entitled 'Effect of antioxidant supplementation on milk yield and quality in Italian lactating buffaloes' is a field study that aimed to verify whether the supplementation of antioxidants (SOD, Zn and Se) in the diet of lactating buffaloes could influence feed intake, feeding behaviour, feeding efficiency, milk yield and milk quality.

The experiment was conducted on sixty-six lactating buffaloes (MI) divided into two groups (33 treated + 33 control) reared on a farm located in central Italy.

Individual milk samples were collected and analysed monthly, while unifeed samples were analysed every fortnight. The results showed that the treatment had no effect on feed intake, feeding behaviour and feeding efficiency, while it improved milk yield and milk clotting properties (rennet clotting time, curd firming and curd firmness).

Overall, this is a good example of what I call useful research. The approach used is clearly presented, but some important details are missing in the Materials and Methods section. I recommend improving the section on chemical-physical, cytological and rheological analyses carried out on milk, by better specifying the equipment used and the analytical methods adopted.

In the title and M&M, if the authors agree, it would be appropriate to specify the buffalo breed.

In the conclusions, I suggest including the percentage increase in milk production of the treated group compared to the control group.

Other than that, I have only minor specific comments:

Line 39: check “B6 ,and minerals.”

Line 97: “measure milk yield”, rewrite “measure milk yield (L)”

Line 133-138: Milk samples were refrigerated (4°C) and analysed within 24 to 36 h after collection. Content (%) of fat, protein, lactose and solid not fat (SNF), and pH were determined by I.R. spectrophotometry, and somatic cell count (SCC, thousand/cells/mL) by fluoro-optoelectronics (MilkoScanTM 7 RM, FOSS, Foss Allé 1 DK-3400 Hilleroed, Denmark). Milk clotting properties (MCP) [RCT (rennet  coagulation time, min); k20 (curd firming time, min); a30 (curd firmness, mm)] were determined by lactodynamography (Mape System, Firenze, Italy) [17].

Specify in detail the equipment used and the analytical methods adopted.

Line 172-181: In paragraph "2.3. Statistical analysis", it would be appropriate to briefly specify the expression of the results,; e.g. results are expressed as lsmeans ...

Line 185, 186 and other parts of the text: '(mean±DS)', probably typos (mean±DS); same consideration for “lsmeans”.

Line 230: in table 5, rewrite "SCC, thousand cells/ml".

Line 270-273: Moreover, Kantwa et al. [31] and Tanwar et al. [32] reported a significant increase in milk yield of buffaloes supplemented with chelated mineral mixture. Other studies have reported an increase of milk yield in dairy cows supplied with organic mineral mixture [33, 34], and area specific mineral mixture [35]. I suggest to Authors of including, if available, the increase in milk yield reported in the references.

Line 315-317: Somatic cells count wa lower than values reported by Tripaldi et al. [40] (314x103 cells/mL) and by Pasquini et al. [43] for buffaloes bred in Marche region (from 152.84±25.22 to 199.73±23.43 x103 cells/mL) and was higher compared with mid-lactating Marrah Buffaloes in India [29].  I suggest including a brief comment on the different analytical methods used (opto-fluoro-electronic vs. CMT).

Line 318: “Marrah”; rewrite in Murrah

Line 340: chek “Infact”

Line 383: 'The treatment positively improved milk production', I suggest inserting the '%' or 'the ratio of means' for milk production found in the two groups.

Author Response

We thank the reviewer for her/his effort in giving suggestions and comments that were useful for improving the manuscript.

R2

The article entitled 'Effect of antioxidant supplementation on milk yield and quality in Italian lactating buffaloes' is a field study that aimed to verify whether the supplementation of antioxidants (SOD, Zn and Se) in the diet of lactating buffaloes could influence feed intake, feeding behaviour, feeding efficiency, milk yield and milk quality. The experiment was conducted on sixty-six lactating buffaloes (MI) divided into two groups (33 treated + 33 control) reared on a farm located in central Italy. Individual milk samples were collected and analysed monthly, while unifeed samples were analysed every fortnight. The results showed that the treatment had no effect on feed intake, feeding behaviour and feeding efficiency, while it improved milk yield and milk clotting properties (rennet clotting time, curd firming and curd firmness).

R2: Overall, this is a good example of what I call useful research. The approach used is clearly presented, but some important details are missing in the Materials and Methods section. I recommend improving the section on chemical-physical, cytological, and rheological analyses carried out on milk, by better specifying the equipment used and the analytical methods adopted.

AU: Thanks for the comments. The materials and methods have been improved as requested.

R2: In the title and M&M, if the authors agree, it would be appropriate to specify the buffalo breed.

AU: done

R2: In the conclusions, I suggest including the percentage increase in milk production of the treated group compared to the control group.

AU: done

Other than that, I have only minor specific comments:

R2: Line 39: check “B6, and minerals.”

AU: done

R2: Line 97: “measure milk yield”, rewrite “measure milk yield (L)”

AU: done

R2: Line 133-138: Milk samples were refrigerated (4°C) and analysed within 24 to 36 h after collection. Content (%) of fat, protein, lactose and solid not fat (SNF), and pH were determined by I.R. spectrophotometry, and somatic cell count (SCC, thousand/cells/mL) by fluoro-optoelectronics (MilkoScanTM 7 RM, FOSS, Foss Allé 1 DK-3400 Hilleroed, Denmark). Milk clotting properties (MCP) [RCT (rennet  coagulation time, min); k20 (curd firming time, min); a30 (curd firmness, mm)] were determined by lactodynamography (Mape System, Firenze, Italy) [17].

Specify in detail the equipment used and the analytical methods adopted.

 AU: done

R2: Line 172-181: In paragraph "2.3. Statistical analysis", it would be appropriate to briefly specify the expression of the results,; e.g. results are expressed as lsmeans ...

AU: done

R2: Line 185, 186 and other parts of the text: '(mean±DS)', probably typos (mean±DS); same consideration for “lsmeans”.

AU: Done. Thanks for the comment. The reviewer is right, we made changes revising the terms.

R2: Line 230: in table 5, rewrite "SCC, thousand cells/ml".

AU: Done.

R2: Line 270-273: Moreover, Kantwa et al. [31] and Tanwar et al. [32] reported a significant increase in milk yield of buffaloes supplemented with chelated mineral mixture. Other studies have reported an increase of milk yield in dairy cows supplied with organic mineral mixture [33, 34], and area specific mineral mixture [35]. I suggest to Authors of including, if available, the increase in milk yield reported in the references.

AU: If the reviewer agrees, we think it is not necessary to report the increase of milk yield found in the literature cited.

R2: Line 315-317: Somatic cells count was lower than values reported by Tripaldi et al. [40] (314x103 cells/mL) and by Pasquini et al. [43] for buffaloes bred in Marche region (from 152.84±25.22 to 199.73±23.43 x103 cells/mL) and was higher compared with mid-lactating Murrah Buffaloes in India [29].

I suggest including a brief comment on the different analytical methods used (opto-fluoro-electronic vs. CMT).

AU: Done.

R2: Line 318: “Marrah”; rewrite in Murrah

AU: Done

R2: Line 340: chek “Infact”

AU: Done

R2: Line 383: 'The treatment positively improved milk production', I suggest inserting the '%' or 'the ratio of means' for milk production found in the two groups.

AU: Done
